# C-Reactive Protein and White Blood Cell Count in Non-Infective Acute Ischemic Stroke Patients Treated with Intravenous Thrombolysis

**DOI:** 10.3390/jcm10081610

**Published:** 2021-04-10

**Authors:** Marcin Wnuk, Justyna Derbisz, Leszek Drabik, Agnieszka Slowik

**Affiliations:** 1Department of Neurology, Jagiellonian University Medical College, 31-688 Krakow, Poland; justyna.derbisz@gmail.com (J.D.); slowik@neuro.cm-uj.krakow.pl (A.S.); 2University Hospital in Krakow, 30-688 Krakow, Poland; 3Department of Pharmacology, Jagiellonian University Medical College, 31-531 Krakow, Poland; leszek.drabik@uj.edu.pl; 4John Paul II Hospital, Krakow, 31-202 Krakow, Poland

**Keywords:** stroke, thrombolysis, C-reactive protein, white blood cell count, prognosis, outcome

## Abstract

**Background:** Previous studies on inflammatory biomarkers in acute ischemic stroke (AIS) produced divergent results. We evaluated whether C-reactive protein (CRP) and white blood cell count (WBC) measured fasting 12–24 h after intravenous thrombolysis (IVT) were associated with outcome in AIS patients without concomitant infection. **Methods:** The study included 352 AIS patients treated with IVT. Excluded were patients with community-acquired or nosocomial infection. Outcome was measured on discharge and 90 days after stroke onset with the modified Rankin scale (mRS) and defined as poor outcome (mRS 3–6) or death (mRS = 6). **Results:** Final analysis included 158 patients (median age 72 years (interquartile range 63-82), 53.2% (*n* = 84) women). Poor outcome on discharge and at day 90 was 3.8-fold and 5.8-fold higher for patients with CRP ≥ 8.65 mg/L (fifth quintile of CRP), respectively, compared with first quintile (<1.71 mg/L). These results remained significant after adjustment for potential confounders (odds ratio (OR) on discharge = 10.68, 95% CI: 2.54–44.83, OR at day 90 after stroke = 7.21, 95% CI: 1.44–36.00). In-hospital death was 6.3-fold higher for patients with fifth quintile of CRP as compared with first quintile and remained independent from other variables (OR = 4.79, 95% CI: 1.29–17.88). Independent predictors of 90-day mortality were WBC < 6.4 × 10^9^ /L (OR = 5.00, 95% CI: 1.49–16.78), baseline National Institute of Health Stroke Scale (NIHSS) score (OR = 1.13 per point, 95% CI: 1.01–1.25) and bleeding brain complications (OR = 5.53, 95% CI: 1.59–19.25) but not CRP ≥ 8.65 mg/L. **Conclusions:** Non-infective CRP levels are an independent risk factor for poor short- and long-term outcomes and in-hospital mortality in AIS patients treated with IVT. Decreased WBC but not CRP is a predictor for 90-day mortality.

## 1. Introduction

Numerous biomarkers [1,2], including inflammatory ones, were found to be associated with atherosclerosis and cardiovascular events [3]. This applied to the levels of C-reactive protein (CRP) which increase during the first 24 h after hospital admission was independently associated with an increased risk of 30-day mortality in patients with acute myocardial infarction [4]. Increased white blood cell count (WBC) was instead an independent risk factor for long-term mortality in patients with coronary artery disease [5].

In patients with acute ischemic stroke (AIS) treated with intravenous thrombolysis (IVT), CRP within 24 h from symptom onset [6] and WBC within 24 h after IVT [7] were associated with a poor long-term functional outcome. However, there were also numerous studies which produced divergent results concerning the prognostic role of CRP in patients with AIS, likely due to a high proportion of patients with an infection. The negative prognostic influence of CRP was not ameliorated by IVT [8], whereas in several studies other prognostic factors were found to be more important in long-term prognosis [9,10,11]. It was shown that from 18.2 [12] to 23.6% [13] of AIS patients developed an infection during hospitalization, with pneumonia and urinary tract infection being the most common [14]. Although the elevation of inflammatory markers is usually observed in infections, a CRP increase in AIS may reflect non-infective ischemia-induced inflammation contributing to a hypercoagulable state and extensive tissue damage [15]. There is no widely accepted cut-off point in the literature for infective CRP in the AIS studies [9,16]. Although the specific cut-off value (>6 mg/dL) was used to exclude possible concomitant infection in a study by Montaner et al. [8], the clinical assessment plays a more important role in the exclusion of infection. Therefore, previous studies might have produced inconsistent findings of the prognostic role of CRP in AIS patients due to the variable and inadequate threshold of a biomarker used. CRP level and WBC may also be affected by measurements which took place in different clinical conditions, such as different time of the day, after previous food intake and under the influence of non-steroidal anti-inflammatory drugs [17,18,19]. For example, fasting is associated with a significant decrease in the CRP level [17], whereas WBC increases by nearly 10% two hours after meal consumption [19].

Many factors, including age, comorbidities and severity as measured with the National Institute of Health Stroke Scale (NIHSS) affect long-term prognosis after AIS treated with IVT [20]. There is still a need for other markers which could help predict long-term prognosis among AIS patients.

Therefore, the aim of the current study was to evaluate whether CRP levels and WBC measured fasting 12–24 h after IVT were associated with short- and long-term outcome in AIS patients without concomitant infection.

## 2. Materials and Methods

### 2.1. Patient Recruitment, CRP and WBC Measurements

The study was performed as a retrospective analysis of the prospectively collected data of 352 AIS patients from the Krakow Stroke Data Bank, a single-center stroke registry established in the University Hospital in Krakow in 2007. All patients were of Caucasian origin and were treated with IVT between June 2014 and December 2018. Excluded were patients with infection (*n* = 83) as described previously [13]. The lack of infection was determined by the exclusion of patients with fever, signs of infection in the physical examination, or those who needed antibiotics during subsequent hospitalization. Moreover, all patients underwent a routine chest x-ray, urine test and internal medicine consultation during the first 3 days since admission. Additionally, follow-up internal medicine consultation was performed in case of CRP elevation and before discharge. As the presence of malignant tumor was the exclusion criteria for IVT in AIS, there were no patients with this condition enrolled in our study.

After further exclusion of patients without available information regarding CRP and WBC levels measured fasting between 12 and 24 h after IVT, there were 158 remaining patients included in the final analysis. Our institutional protocol for fasting blood measurements requires overnight fasting for at least 6 h [21] and blood withdrawal before breakfast between 7 and 8 a.m. [22].

We collected the data about demographics, vascular risk factors, stroke etiology and NIHSS on admission and after IVT. Patients were followed-up according to the previously described protocol [13]. Outcome was measured with the modified Rankin scale (mRS). We obtained information about prognosis on discharge and 90 days after stroke onset and defined as poor outcome (mRS 3–6) or death (mRS = 6). Bleeding brain complications secondary to IVT were defined according to the ECASS-1 classification [23,24].

The study was approved by the Jagiellonian University Ethical Committee (KBET 54/B/2007). We obtained informed consent from all patients which was either written or verbal in the presence of at least two physicians in case of the inability to use the dominant hand due to stroke.

### 2.2. Statistics

The study was powered to have an 80% chance to detect a 50% difference in CRP between the group with mRS 0–2 and 3–6 at the 0.05 significance level. In order to demonstrate such a difference or greater, 26 subjects or more were required in each group based on the values from a published article [16].

The baseline clinical characteristics according to the quintiles of CRP and quartiles of WBC were compared using the univariate analysis of variance, Kruskal–Wallis rank ANOVA and chi-squared test, as appropriate. Values were presented as counts and percentages, means and standard deviations or medians (interquartile ranges), as appropriate. The multivariate models included age, sex and body mass index (BMI) (model 1); age, sex, BMI, hypertension, mRS score before stroke > 0, baseline NIHSS score, hemorrhagic brain complications and fasting hyperglycemia (model 2A); and age, sex, BMI, hypertension, maximal systolic blood pressure within 24 h after IVT, mRS score before stroke > 0, baseline NIHSS score, hemorrhagic brain complications and mechanical thrombectomy (model 2B). The Hosmer–Lemeshow test and ROC (Receiver Operating Characteristic) scores were used to evaluate the goodness of fit. Models with the lowest Akaike information criterion and highest Nagelkerke pseudo R2 were presented. The level of significance was set at a *p*-value ≤ 0.05. The data were processed using STATISTICA version 13.0 (Statsoft Inc., Tulsa, OK, USA).

## 3. Results

### 3.1. Patient Characteristics, Serum CRP Levels and WBC 12–24 h after IVT

The cohort of 158 patients with AIS treated with IVT is presented in Table 1 according to the CRP quintiles. The median age was 72 years (interquartile range 63–82), and 53.2% (*n* = 84) of patients were women. Mechanical thrombectomy was performed in 47 (29.7%) patients.

The median CRP 12–24 h after IVT was 3.94 (2.01–5.12 mg/L) and 15.2% (*n* = 24) of patients had CRP > 10 mg/L. The BMI, NIHSS score on admission and after IVT and prevalence of hypertension increased with the increase in CRP. The median WBC was 7.76 (6.40–9.61 × 10^9^/L). Large-vessel disease stroke was associated with a higher WBC (Appendix A). There was no correlation between CRP and fasting glucose levels (r = 0.11, *p* = 0.167).

### 3.2. Serum CRP Levels and WBC 12–24 h after IVT and Poor Functional Outcome on Discharge and at Day 90 after Stroke

Patients with poor functional outcome on discharge (*n* = 52, 32.9%) and at day 90 after stroke (*n* = 39, 25.3%) had higher CRP measured fasting between 12 and 24 h after IVT compared with the remainder (5.92 (3.31–12.10) vs. 3.16 (1.76–5.73) mg/L, *p* < 0.001 and 5.91 (2.87–11.27) vs. 3.43 (1.71–5.96) mg/L, *p* < 0.001, respectively). The area under the receiver operating characteristic curve (AUC) of CRP for predicting poor functional outcome at day 90 was 0.667 (95% CI, 0.559–0.773, *p* = 0.002). With a cut-off point of 8.65 mg/L, the sensitivity, specificity and accuracy were 43.6%, 89.6% and 77.9%, respectively. The frequency of poor functional outcome on discharge and at day 90 was 3.8-fold and 5.8-fold higher for patients with CRP ≥ 8.65 mg/L (the fifth quintile of CRP), respectively, compared with the first quintile (<1.71 mg/L) (Figure 1). The odds ratio of poor functional outcome for patients in the highest quintile of CRP on discharge remained significant after adjustment for age, sex, BMI, hypertension, mRS score before stroke >0, baseline NIHSS score, hemorrhagic brain complications and fasting hyperglycemia (Table 2). The association between CRP and poor functional outcome at day 90 after stroke was significant in the multivariable model adjusted for multiple confounders including age, sex, baseline NIHSS score, bleeding brain complications and mechanical thrombectomy (Table 2). The CRP ≥ 8.65 mg/L was predicted by the NIHSS score on admission and hypertension (Appendix A).

There was no significant association between the quartiles of WBC and the frequency of poor functional outcome on discharge and at day 90 (Figure 1).

### 3.3. Serum CRP Levels and WBC 12–24 h after IVT and Early Neurological Deterioration and Bleeding

Neurological deterioration after IVT, defined as an increase of two or more points in the NIHSS scale [25] between admission and 12–24 h after IVT, was observed in seven (4.43%) patients. The CRP and WBC values did not differ between groups with and without neurological deterioration (5.77 (3.45–9.31) vs. 3.79 (1.03–7.36) mg/L, *p* = 0.22 and 10.42 (4.95–13.95) vs. 7.77 (6.44–9.77) × 10^9^/L, *p* = 0.29, respectively). Intracranial bleeding (ECASS class 1–4) was observed in 28 (17.72%) patients (6.96%, 5.06%, 3.16% and 2.53% for classes 1–4, respectively). We observed a trend toward higher CRP (5.18 (2.99–9.44) vs. 3.56 (1.84–7.34) mg/L), *p* = 0.09), and no difference in WBC (7.12 (6.11–10.38) vs. 7.86 (6.51–9.45) × 10^9^/L, *p* = 0.43), in patients with and without bleeding, ECASS class 1–4.

### 3.4. Serum CRP Levels and WBC 12–24 h after IVT and In-Hospital and 90-Day Mortality

Patients who died in hospital (*n* = 11, 7.0%) and within 90 days after stroke (*n* = 18, 11.4%) had higher CRP levels compared with the survivors (8.99 (4.39–15.42) vs. 3.67 (1.93–6.93) mg/L, *p* = 0.01 and 6.51 (4.39–10.67) vs. 3.49 (1.83–6.23) mg/L, *p* = 0.006, respectively). The frequency of in-hospital death was 6.3-fold higher for patients with the fifth quintile of CRP as compared with the first quintile (Figure 1). In-hospital mortality was independently predicted by CRP ≥ 8.65 mg/L and hemorrhagic brain complications (Table 3). The independent predictors of 90-day mortality were WBC < 6.4 × 10^9^/L, baseline NIHSS score and hemorrhagic brain complications, but not CRP ≥ 8.65 mg/L (Table 3).

The AUC of WBC for 90-day mortality was 0.660 (95% CI, 0.518–0.802, *p* = 0.027). The WBC value of 6.4 × 10^9^/L offered the best overall sensitivity, specificity and accuracy of 76.5, 23.5 and 74.0%, respectively. The WBC < 6.4 × 10^9^/L was associated with a 4.7-fold and 8.8-fold higher rate of deaths compared with the WBC 6.40–7.75 × 10^9^/L and 7.76–9.60 × 10^9^/L (Figure 1). The WBC < 6.4 × 10^9^/L was predicted by large vessel stroke etiology (Appendix A).

## 4. Discussion

The current study supported an association between fasting CRP levels measured 12–24 h after IVT and analyzed in quintiles and long-term functional outcome after AIS in patients without concomitant infection. Previous prospective study performed on a large cohort of more than 3000 patients confirmed the prognostic significance of CRP measured within 24 h from symptom onset in the whole group of patients assessed with mRS three months after AIS, and, admittedly, after adjustment for IVT, this association was still maintained [6]. A similar conclusion came from the study evaluating 436 AIS patients without infection in China, in which elevated CRP levels within 24 h after IVT were found to increase nearly 5-fold the risk of 3-month poor functional outcome [7]. On the other hand, in the study of Karlinski et al., also evaluating AIS patients without infection, CRP assessed within 24 h from symptom onset was not associated with 3-month outcome, however, patients were dichotomized according to the abnormal level of CRP, that is, lower than or above 5 ng/mL [16]. In another study, the change in CRP between admission and the seventh day of hospitalization did not affect long-term outcome either [11]. However, the long-term prognostic role of CRP in AIS patients without infection treated with IVT seemed to be also supported by the studies in patients undergoing another reperfusion therapy of AIS, that is, mechanical thrombectomy. In a recent study, a similar to our research cut-off level of CRP was found to be associated with long-term outcome in AIS patients who underwent endovascular therapy independent of other variables [26].

In our study, no association between CRP levels and 3-month mortality after AIS treated with IVT was found. Our conclusions stayed in line with the previous results coming from the Chinese study which also did not support a correlation of CRP with all-cause mortality at 3 months [7]. Interestingly, in the same study, high WBC increased 2-fold the risk of death at 3 months [7]. Increased WBC the next day after mechanical thrombectomy was found to significantly correlate with the NIHSS score at day 90 after stroke onset [27]. We found instead that the lowest WBC quartile was an independent risk factor for increased long-term mortality at 3 months in AIS patients without infection. Together with the observed trend for higher mortality in the quartile of the highest WBC, it seemed that the association between WBC and the risk of death might resemble the U-curve.

In our specific group of patients, the association of CRP with short-term outcome was also revealed. Our observation resembled the results from another Chinese study which searched for the predictors of poor response to IVT [28]. The authors proposed even the ACBS scale, as there were four parameters which remained significant in the multivariate analysis, with CRP apart from age, glucose levels and systolic blood pressure at baseline [28]. Elevated CRP levels were also found to correlate negatively with the reduction in neurological deficit measured with the NIHSS 12–24 h after IVT [29]. Thus, it seems that assessment of CRP within the first 24 h after admission in AIS patients without infection treated with IVT could be helpful in the evaluation of their short-term clinical outcome. However, uncertainty exists concerning the appropriate time point for CRP measurement in AIS patients. Most previous studies assessed CRP levels within 24 h after admission [16], whereas in others measurements took place within 24 h from AIS onset [6] or upon arrival to the emergency department [28]. In our study, we used the specific time frame for CRP assessment, that is, between 12 and 24 h after IVT, which was previously used in another study evaluating the prognostic role of trends in CRP levels [29]. Another study on a group of more than a thousand AIS patients receiving IVT showed that CRP levels assessed within 24 h from admission or in the following days better predicted long-term functional outcome that the admission values [30]. It is worth mentioning that in reference to another blood parameter such as glucose, its values measured fasting the next morning after admission had more potent long-term prognostic significance than the admission values [22]. Therefore, as levels of CRP might be biased by food intake [17], we measured CRP levels in our AIS patients after at least 6 h overnight fasting [21], according to our institutional protocol [22]

There were no associations between CRP levels and neurological worsening or bleeding brain complications in our study. Previous studies showed that patients with elevated CRP more often suffered from bleeding brain complications, however, this association was not confirmed after adjustment for other clinical parameters, including age, stroke severity and recent infection [16], which resembles the results of our study. As to neurological deterioration, higher CRP levels were found to be associated with its increased risk in another study after adjustment for multiple confounders [6] that was not confirmed in our study, probably due to the smaller sample size. In another study, the association between CRP and neurological deterioration was supported only for AIS patients who did not receive IVT [31].

The role of CRP in the pathophysiology of stroke is complicated, however, during AIS, systemic inflammation is induced, which may result in increased body temperature, WBC and CRP levels [32]. In a rabbit model of stroke, CRP levels after a cerebrovascular event correlated with the size of the infarct, and were perceived by the authors as a good marker of prognosis during AIS assessment [33]. Early treatment with IVT may result in the reduction in inflammatory response due to inhibition of the brain tissue necrosis [32].

In our study, CRP after adjustment for numerous variables including hypertension predicted poor functional outcomes. Interestingly, we observed that hypertension was the strongest predictor of the highest levels of CRP in the multivariate analysis (Appendix A). We hypothesize that elevated CRP may reflect the risk of an unfavorable outcome that was attributed to hypertension in other studies [34]. Future studies are needed to evaluate those mechanisms. Although there was no statistically significant correlation between CRP and fasting glucose levels, we cannot exclude the true correlation with the increase in the sample size.

Non-infective CRP in patients with AIS may deliver prognostic information distinct from that carried by CRP measured during the acute phase of infection. Non-infective CRP elevation may reflect stroke-induced inflammation and endothelial dysfunction contributing to a hypercoagulable state and extensive tissue damage. High levels of CRP are an independent marker of cardiovascular risk, which may be reduced by a statin therapy independently to a lipid-lowering drug action [35]. Aimed anti-inflammatory treatment targeting interleukin-1, endothelial selectins and leukocyte infiltration shows promising results in preclinical and small clinical studies in AIS [36]. Therefore, we view the results of our study as hypothesis-generating, whereas future studies are needed to evaluate whether such treatment could be effective in diminishing inflammatory processes induced by AIS and manifested by non-infective CRP increase. Finally, the unique timing of sample collections that takes into consideration the CRP-kinetics may contribute to our observations.

Our study has important limitations. First, the sample of patients was relatively small and therefore subgroup analyses, especially related to WBC, should be interpreted with caution. Second, we did not analyze the influence of change in CRP levels during hospitalization on long-term functional outcome. Third, we also did not take into account the differential of WBC which was recently shown to be an important prognostic factor in AIS patients [37]. Fourth, the reported statistical associations do not necessarily mean a cause-effect relationship.

## 5. Conclusions

In conclusion, CRP levels measured fasting between 12 and 24 h after IVT are an independent risk factor for poor short- and long-term outcomes in AIS patients without infection, as well as for in-hospital mortality. The lowest quartile of WBC predicts 3-month mortality; however, this finding needs further investigation. Future studies are also expected to create a prognostic scale for short- and long-term prognosis after AIS with CRP as one of the reasonable parameters.

## Figures and Tables

**Figure 1 jcm-10-01610-f001:**
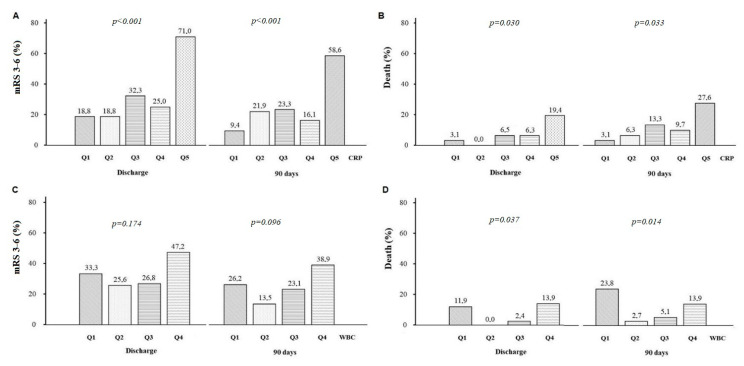
Panel (**A**,**B**). The proportion of patients with disability (modified Rankin Scale, mRS 3–6) or death (mRS = 6) on discharge and at day 90 after stroke onset, according to the quintile (quintile 1–quintile 5, Q1–Q5) of C-reactive protein (CRP) values. Panel (**C**,**D**). The proportion of patients with disability or death on discharge and at day 90 after stroke onset, according to the quartile (quartile 1–quartile 4, Q1–Q4) of white blood cell count (WBC).

**Table 1 jcm-10-01610-t001:** Baseline clinical characteristics of the patients based on C-reactive protein (CRP) quintiles.

	Q1 CRP < 1.71 (*n* = 32)	Q2 CRP 1.71–3.11 (*n* = 32)	Q3 CRP 3.12–5.09 (*n* = 31)	Q4 CRP 5.10–8.64 (*n* = 32)	Q5 CRP ≥ 8.65 (*n* = 31)	*p*-Value
Age (years)	68 (58–76)	73 (68–80)	73 (69–80)	72 (64–83)	77 (62–86)	0.246
Women, *n* (%)	15 (46.9)	15 (46.9)	14 (45.2)	19 (59.4)	21 (67.8)	0.292
BMI (kg/m^2^)	24.7 (22.8–27.7)	24.8 (23.3–27.7)	27.5 (24.2–29.3)	27.1 (25.2–29.9)	26.8 (25.0–29.7)	0.006
Hypertension, *n* (%)	22 (68.8)	23 (71.2)	27 (87.1)	26 (81.2)	30 (96.8)	0.031
Hypercholesterolemia, *n* (%)	12 (37.5)	8 (25.0)	11 (35.5)	11 34.4)	6 (19.4)	0.461
Diabetes mellitus, *n* (%)	9 (28.1)	8 (25.0)	7 (22.6)	7 (21.9)	11 (35.5)	0.744
Smoking, *n* (%)	6 (18.8)	2 (6.3)	2 (6.5)	8 (25.8)	5 (16.1)	0.151
Ischemic heart disease, *n* (%)	7 (21.9)	3 (9.4)	7 (22.6)	10 (31.2)	8 (25.8)	0.307
Atrial fibrillation, *n* (%)	6 (18.8)	10 (31.3)	11 (35.5)	8 (25.0)	9 (29.0)	0.635
Previous stroke, *n* (%)	7 (21.9)	3 (9.4)	8 (25.8)	6 (18.8)	4 (12.9)	0.432
mRS score before stroke >0	3 (9.4)	1 (3.1)	1 (3.2)	4 (12.5)	2 (6.4)	0.523
Stroke etiology, *n* (%) -large-vessel disease -small-vessel disease -cardioembolic -other - undetermined	5 (15.6) 1 (3.1) 6 (18.8) 19 (59.4) 1 (3.1)	5 (15.6) 1 (3.1) 11 (34.4) 14 (43.8) 1 (3.1)	5 (16.1) 0 (0.0) 12 (38.7) 13 (41.9) 1 (3.2)	3 (9.4) 0 (0.0) 11 (34.4) 18 (56.3) 0 (0.0)	5 (16.1) 0 (0.0) 10 (32.3) 15 (48.4) 1 (3.2)	0.932
Mechanical thrombectomy, *n* (%)	5 (15.60	8 (25.0)	10 (32.3)	11 (34.4)	13 (41.9)	0.198
Time from stroke onset to thrombolysis (min)	119 (94–175)	145 (93–174)	117 (85–174)	90 (76–176)	121 (83–160)	0.556
NIHSS score on admission	9.7 ± 6.3	9.2 ± 6.7	12.5 ± 6.9	11.6 ± 6.3	14.6 ± 6.4	0.013
NIHSS score after r-tPA	4.4 ± 4.6	5.3 ± 5.5	8.1 ± 9.7	5.7 ± 6.9	11.9 ± 8.7	<0.001
Post-IVT hemorrhagic brain complications, *n* (%) -no complication -HI type 1 -HI type 2 -PH type 1 -PH type 2	29 (90.6) 1 (3.1) 1 (3.1) 1 (3.1) 0 (0.0)	27 (84.4) 3 (9.4) 1 (3.1) 1 93.1) 0 (0.0)	25 (80.7) 2 (6.5) 1 (3.2) 2 (6.5) 1 (3.2)	27 (84.4) 2 (6.3) 2 (6.3) 0 (0.0) 1 (3.1)	22 (71.0) 3 (9.7) 3 (9.7) 1 (3.2) 2 96.5)	0.874
Maximal SBP within 24 h after r-tPA (mmHg)	144 (123–156)	145 (125–160)	145 (134–166)	151 (139–169)	143 (134–160)	0.536
Maximal DBP within 24 h after r-tPA (mmHg)	80 (72–90)	80 (72–89)	80 (70–90)	80 (70–85)	77 (70–80)	0.694
Fasting glucose (mmol/L)	6.5 (5.5–7.3)	6.2 (5.4–7.0)	6.1 (5.4–8.2)	6.8 (5.5–7.9)	6.9 (5.9–8.2)	0.491
Creatinine (µmol/L)	82 (65–94)	74 (63–100)	77 (69–97)	82 (71–91)	76 (65–93)	0.892
WBC (×10^9^/L)	7.1 (6.0–8.7)	7.8 (6.6–8.8)	7.5 (5.4–9.2)	8.2 (6.9–10.9)	8.9 (6.4–11.5)	0.145

Values are presented as *n* (%), mean ± standard deviation, median and interquartile range. Abbreviations: BMI—body mass index, DBP—diastolic blood pressure, IVT—intravenous thrombolysis, HI—hemorrhagic infarction, mRS—modified Rankin scale, MT—mechanical thrombectomy, NIHSS—National Institutes of Health Stroke Scale, PH—parenchymal hematoma, r-tPA—recombinant tissue plasminogen activator, SBP—systolic blood pressure, and WBC—white blood cells count. Q1–Q5 denotes five groups according to the quintile of CRP (mg/L).

**Table 2 jcm-10-01610-t002:** C-reactive protein and poor functional outcome (mRS 3–6) on discharge and at day 90 after stroke.

On Discharge		Model 1		Model 2A	
	Events *n*, (%)	OR (95% CI)	*p*-value	OR (95% CI)	*p*-value
Q1 (<1.71, *n* = 32)	6 (18.8)	1.00 (reference)	-	1.00 (reference)	-
Q2 (1.71–3.11, *n* = 32)	6 (18.8)	1.00 (0.28–3.56)	1.00	1.28 (0.27–5.96)	0.756
Q3 (3.12–5.09, *n* = 31)	10 (32.3)	2.13 (0.66–6.94)	0.209	1.65 (0.40–6.86)	0.493
Q4 (5.10–8.64, *n* = 32)	8 (25.0)	1.33 (0.40–4.45)	0.647	1.02 (0.24–4.29)	0.982
Q5 (≥8.65, *n* = 31)	22 (71.0)	9.70 (2.94–31.98)	<0.001	10.68 (2.54–44.83)	0.001
P for trend			<0.001		0.004
AIC			182.19		132.89
AUC			0.744 ± 0.05		0.895 *±* 0.03
R2 Nagelkerke			0.241		0.560
Hosmer–Lemeshow test *p*-value			0.369		0.057
At day 90 after stroke		Model 1		Model 2B	
	Events *n*, (%)	OR (95% CI)	*p*-value	OR (95% CI)	*p*-value
Q1 (<1.71, *n* = 32)	3 (9.4)	1.00 (reference)	-	1.00 (reference)	-
Q2 (1.71–3.11, *n* = 32)	7 (21.9)	2.90 (0.65–13.0)	0.165	4.11 (0.73–23.7)	0.109
Q3 (3.12–5.09, *n* = 31)	7 (22.5)	3.18 (0.71–14.4)	0.132	1.86 (0.33–10.63)	0.484
Q4 (5.10–8.64, *n* = 32)	5 (15.7)	1.64 (0.34–7.83)	0.535	0.75 (0.12–4.53)	0.754
Q5 (≥ 8.65, *n* = 31)	17 (54.8)	12.56 (2.95–53.5)	0.001	7.21 (1.44–36.0)	0.016
P for trend			0.022		0.021
AIC			152.42		125.16
AUC			0.776 ± 0.04		0.887 ± 0.03
R2 Nagelkerke			0.291		0.521
Hosmer–Lemeshow test *p*-value			0.169		0.569

Model 1 included age, sex and BMI. Model 2A included age, sex, BMI, hypertension, mRS score before stroke >0, baseline NIHSS score, hemorrhagic brain complications and fasting hyperglycemia. Model 2B included age, sex, BMI, hypertension, maximal SBP within 24 h after r-tPA, mRS score before stroke >0, baseline NIHSS score, hemorrhagic brain complications and mechanical thrombectomy. Abbreviations, see Table 1. AIC denotes Akaike information criterion, AUC—the area under the curve, CI—confidence interval, OR—odds ratio, Q1–Q5 five groups according to the quintile of CRP (mg/L).

**Table 3 jcm-10-01610-t003:** Predictors of death on discharge and at day 90 after stroke.

On Discharge	OR	95% CI	*p*-Value	OR	95% CI	*p*-Value
Age (per 1 year)	1.00	0.96–10.5	0.861	-	-	-
Sex (female)	4.32	0.90–20.68	0.067	-	-	-
BMI (per 1 unit)	1.05	0.92–1.21	0.441	-	-	-
Baseline NIHSS score (per 1 point)	1.15	1.03–1.28	0.012	-	-	-
Mechanical thrombectomy	3.10	0.90–10.73	0.074	-	-	-
Hemorrhagic brain complications (ECASS 1–3 score)	6.82	1.91–24.28	0.003	5.66	1.52–21.11	0.010
CRP ≥ 8.65 mg/L	5.86	1.66–20.70	0.006	4.79	1.29–17.88	0.020
WBC < 6.4 ×10^9^/L	2.48	0.71–8.59	0.152	-	-	-
AIC AUC						72.21 0.794 ± 0.07
R2 Nagelkerke						0.208
Hosmer–Lemeshow test *p*-value						0.689
At day 90 after stroke	OR	95% CI	*p*-value	OR	95% CI	*p*-value
Age (per 1 year)	1.03	0.99–1.08	0.161	-	-	-
Sex (female)	3.50	1.10–11.18	0.034	-	-	-
BMI (per 1 unit)	1.04	0.93–1.16	0.523	-	-	-
Baseline NIHSS score (per 1 point)	1.17	1.07–1.28	<0.001	1.13	1.01–1.25	0.036
Hemorrhagic brain complications (ECASS 1–3 score)	8.19	2.86–23.50	<0.001	5.53	1.59–19.25	0.007
CRP ≥ 8.65 mg/L	4.38	1.55–12.39	0.005	-	-	-
WBC < 6.4 ×10^9^/L	4.06	1.48–11.16	0.007	5.00	1.49–16.78	0.009
AIC AUC						87.03 0.877 ± 0.04
R2 Nagelkerke						0.406
Hosmer–Lemeshow test *p*-value						0.420

Abbreviations, see Table 1 and Table 2.

## Data Availability

The data supporting the findings of the present study will be made available for any qualified investigator from the corresponding author upon reasonable request.

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
