# Peer review of "C-Reactive Protein and White Blood Cell Count in Non-Infective Acute Ischemic Stroke Patients Treated with Intravenous Thrombolysis"

_jcm, 2021, doi:10.3390/jcm10081610_

Round 1

Reviewer 1 Report

Thank you very much for giving me an opportunity to review a paper entitled, “C-Reactive Protein and White Blood Cell Count in Non-Infective Acute Ischemic Stroke Patients Treated with Intravenous Thrombolysis.”

Major:

As authors presented, many previous studies demonstrated the association of CRP elevation with clinical outcome after stroke. I read this paper with hope that this paper might have discussed the etiology of CRP elevation. But unfortunately, there is no new take home message.

It is not clear how authors diagnosed enrolled patients as not having infection. It is not easy to deny the infection in clinical practice.

How many patients suffered from malignant tumor?

Reviewer 2 Report

The authors describe that WBC but not CRP is predictor for 90-day mortality in AIS patients treated with IVT in a retrospective study.

  1. It is unclear from the introduction what novelty the current study provides.
  2. The introduction should better funnel towards the actual purpose and knowledge gain of the current study. For instance, it is unclear what % of patients generally have infections and thus, how this affects or explains controversial results from previous studies.
  3. How does fasting affect WBC count and CRP levels?
  4. Is the protocol for fasting standardised to assure comparable results to be used for a retrospective study.
  5. The authors state that "the odds ratio of poor functional outcome for patients in the highest quintile of CRP on discharge remained significant after adjustment for age, sex, BMI, hypertension, mRS score before stroke >0, baseline NIHSS score, hemorrhagic brain complications and fasting hyperglycemia". How do the authors explain the relative independence from hypertension and hyperglycaemia - both with clear inflammatory profiles?  
  6. Similarly, "The role of CRP in the pathophysiology of stroke is complicated, however, during AIS, systemic inflammation is induced..". Most patients have a precondition that would associate to systemic CRP. How would the authors interpret their results in light of this?
  7. Figure 1 is hard to read. Larger panels and larger font size would improve readability.

Reviewer 3 Report

Wnuk et al. conduced a clinical study evaluating whether C-reactive protein and white blood cell count after 12-24 h after intravenous thrombolysis are associated with the outcome in AIS patients. The data are well-organized and in good quality. There are good a few minor suggestions before accepting for publication.

  1. The quartile information should be labeled clearly for easier reading in figure 1.
  2. In table 1 and supplemental table 1, any values that are not represented as percentage should provide as mean+SD and median.

Round 2

Reviewer 1 Report

The information on "the lack of infection" is clearly described properly, which has significantly increased the quality of this paper. There is no further comment. 

Reviewer 2 Report

I have no further comments.